# Prevalence, Management, and Outcomes of Atrial Fibrillation in Paediatric Patients: Insights from a Tertiary Cardiology Centre

**DOI:** 10.3390/medicina60091505

**Published:** 2024-09-15

**Authors:** Andreia Duarte Constante, Joana Suarez, Guilherme Lourenço, Guilherme Portugal, Pedro Silva Cunha, Mário Martins Oliveira, Conceição Trigo, Fátima F. Pinto, Sérgio Laranjo

**Affiliations:** 1Pediatric Cardiology Department, Reference Center for Congenital Heart Diseases, Hospital de Santa Marta, Unidade Local de Saúde São José EPE, 1150-199 Lisbon, Portugal; 2Centro Clínico Académico de Lisboa, Clínica Universitária de Cardiologia Pediátrica, 1169-045 Lisbon, Portugal; 3Arrhythmology, Pacing and Electrophysiology Unit, Cardiology Department, Santa Marta Hospital, Centro Hospitalar Universitário Lisboa Central, 1649-004 Lisbon, Portugal; 4Centro Clínico Académico de Lisboa, Clínica Universitária de Cardiologia, 1169-045 Lisbon, Portugal; 5Instituto de Fisiologia, Faculdade de Medicina, Universidade de Lisboa, 1649-004 Lisbon, Portugal; 6Comprehensive Health Research Center, NOVA Medical School, Faculdade de Ciências Médicas, NMS, FCM, Universidade NOVA de Lisboa, 1169-056 Lisbon, Portugal

**Keywords:** atrial fibrillation, paediatric arrhythmias, paediatric cardiology, congenital heart disease, rheumatic heart disease

## Abstract

*Background and Objectives:* Atrial fibrillation (AF) is increasingly recognised in paediatric patients, presenting unique challenges in management due to its association with various underlying heart conditions. This study aimed to evaluate the prevalence, management strategies, and outcomes of AF in this population. *Materials and Methods*: A retrospective analysis was conducted at a tertiary paediatric cardiology centre, including patients aged ≤18 years diagnosed with AF between January 2015 and December 2023. The study focused on demographic details, clinical presentations, treatments, and outcomes. Descriptive statistics were employed to assess treatment efficacy, recurrence rates, and complications. *Results*: The study included 36 paediatric patients (median age: 15 years, IQR: 13–17; 58% male). Of these, 52.8% had acquired heart disease, 16.7% had congenital heart anomalies, and 16.7% presented with lone AF. The initial management strategies involved electrical cardioversion in 53.3% of patients and pharmacological conversion with amiodarone in 46.7%. Rhythm control therapy was administered to over 80% of the cohort, and 63.9% were placed on oral anticoagulation, predominantly for rheumatic and congenital heart diseases. The overall success rate of rhythm control was 96.2%, with an AF recurrence rate of 3.8%. Ischemic stroke was the most common complication, occurring in three patients, all with underlying rheumatic heart disease. *Conclusions*: AF in paediatric patients is predominantly associated with rheumatic and congenital heart diseases, though a significant proportion of patients present with lone AF. Despite effective rhythm control in most cases, neurological complications, particularly ischemic strokes in patients with underlying heart disease, remain a critical concern. These findings underscore the need for more comprehensive studies to better understand the aetiology, risk factors, and optimal management strategies for paediatric AF.

## 1. Introduction

Atrial fibrillation (AF) is a common cardiac arrhythmia in adults, characterised by disorganised electrical activity in the atria, leading to irregular and often rapid heartbeats. This condition has significant global health implications, contributing to increased morbidity, mortality, and healthcare costs. In adults, the mechanisms behind AF are complex and involve structural changes in the heart, electrophysiological disruptions, and autonomic nervous system influences. Common risk factors include hypertension, coronary artery disease, and valvular heart disease. However, in children, AF presents unique challenges due to its rarity and distinct causes [1].

Traditionally, AF in children is considered extremely rare, often attributed to secondary causes such as congenital heart defects. Recent studies, however, indicate that paediatric AF is more common than previously thought, with a prevalence of less than 0.05% in individuals under 30 years old, particularly among those with existing heart conditions [2]. The limited scope of existing studies on paediatric AF highlights the need for further research to fully understand this condition in young patients.

Children with congenital heart disease (CHD) make up a significant portion of the paediatric AF population. Surgical interventions, especially those involving atrial incisions, increase the risk of AF later in life, making postoperative monitoring essential [3]. Other risk factors include cardiomyopathies and inherited arrhythmia syndromes, which can create conditions in the atria that lead to AF [4].

A subset of children with AF, known as lone atrial fibrillation, presents without any identifiable structural heart disease or secondary causes. This rare form of AF suggests a potential genetic predisposition [5]. Recent evidence points to mutations in genes related to ion channels, gap junctions, and atrial muscle structure as possible contributors to AF in otherwise healthy children [6]. However, the exact genetic mechanisms remain unclear and warrant further study.

Atrial cardiomyopathies, which involve structural and functional changes in the atrial muscle, also play a significant role in the development of AF in children. These changes may be subtle and difficult to detect but significantly increase the risk of AF [7]. Understanding how atrial myopathy interacts with AF in children is key to improving diagnosis and developing targeted treatments.

Despite increasing research on paediatric AF, many questions remain about its causes, clinical progression, and best treatment approaches. The unique physiological and developmental characteristics of children make it difficult to apply adult treatment methods to paediatric patients. This study aimed to fill some of these gaps by analysing cases of paediatric AF from a tertiary paediatric cardiology centre, focusing on the epidemiology, management, and outcomes of the condition.

## 2. Materials and Methods

### 2.1. Study Design and Ethical Considerations

This retrospective study was conducted at a tertiary care centre specialising in paediatric cardiology. The study protocol was approved by the Institutional Review Board (IRB) of Unidade Local de Saúde São José (Ethics Committee approval number 974/2020). Given the retrospective nature of the study, the IRB waived the requirement for informed consent. Obtaining individual consent was deemed impractical, because the study involved the analysis of pre-existing medical records without direct patient interaction. Additionally, all data were anonymised to ensure patient confidentiality, aligning with ethical standards and regulations.

The study adhered to the ethical principles outlined in the Declaration of Helsinki. In compliance with the General Data Protection Regulation (GDPR) and national data protection laws, stringent measures were taken to protect patient privacy. The data were anonymised before analysis, and access was restricted to authorised personnel only. Data handling and storage followed institutional protocols to maintain confidentiality and data security.

### 2.2. Patient Selection

The study population included paediatric patients aged 0–18 years who were diagnosed with atrial fibrillation (AF) between January 2015 and December 2023. Patients were identified through a comprehensive review of electronic health records (EHRs). The inclusion criteria required a confirmed diagnosis of AF, as documented by a 12-lead electrocardiogram (ECG), Holter monitoring, or event monitoring records.

### 2.3. Data Collection and Management

Data were collected using a structured form designed to ensure consistency and accuracy. The following variables were systematically gathered:

Demographic Information: This included the age at diagnosis, sex, ethnicity, and relevant family history. We also documented the presence of associated diagnoses, such as genetic conditions or congenital heart disease (CHD). For patients with CHD, details regarding any surgical interventions, the presence of residual lesions (e.g., valvular regurgitation and atrial dilation), and postoperative outcomes were carefully recorded.

Clinical Presentation: We collected data on the presenting symptoms (e.g., palpitations, dizziness, and syncope); the duration of symptoms prior to diagnosis; and potential triggers for AF (e.g., exercise and infections).

Diagnostic Findings: Diagnostic evaluations included data from an ECG, Holter monitoring, and event monitoring, documenting the type and frequency of arrhythmia episodes. We also included findings from advanced imaging modalities such as echocardiography and cardiac magnetic resonance imaging (MRI). These were used to assess for structural heart disease, with particular attention to residual lesions, valvular dysfunction (e.g., mitral regurgitation), atrial dilation, and both systolic and diastolic ventricular function. Cardiac MRI was further utilised to evaluate myocardial fibrosis through late gadolinium enhancement (LGE).

Laboratory and Imaging Results: Laboratory tests, including electrolytes, thyroid function, and relevant genetic testing, were reviewed to identify potential metabolic or endocrine triggers for AF. Advanced imaging studies, particularly cardiac MRI, were prioritised to assess detailed cardiac anatomy, function, and the presence of myocardial fibrosis, all of which could have implications for AF pathophysiology.

Electrophysiological Study (EPS) Data: For patients who underwent an electrophysiological study (EPS), data were collected on the study’s findings, including the inducibility of arrhythmias, electrophysiological characteristics of the atria, and the outcomes of any ablation procedures.

Treatment Modalities: Information on treatment approaches was documented, covering pharmacological therapies (e.g., antiarrhythmic drugs and anticoagulants), non-pharmacological interventions (e.g., electrical cardioversion and catheter ablation), and surgical treatments. The outcomes of EPS and ablation procedures were specifically noted, along with any complications associated with these interventions.

Clinical Outcomes: Clinical outcomes included the resolution of AF, recurrence rates, and complications such as stroke or heart failure. Long-term follow-up data were collected to evaluate the durability of treatment outcomes and to determine the need for further intervention.

Follow-Up Period: Patients were followed up to monitor the recurrence of atrial fibrillation (AF) and assess the long-term efficacy of rhythm control strategies. The median follow-up time for the cohort was 30 months [IQR: 24–36 months].

### 2.4. Statistical Analysis

Descriptive statistics were used to summarise the demographic and clinical characteristics of the study population. Continuous variables were presented as the mean ± standard deviation (SD) or median with interquartile range (IQR), depending on the distribution of the data. Categorical variables were summarised as frequencies and percentages.

To assess the recurrence rate of atrial fibrillation in our cohort, we systematically recorded all instances of AF recurrence during the follow-up period. The recurrence rate was calculated as the proportion of patients who experienced a recurrence of AF out of the total number of patients in the study. In addition to the recurrence rate calculation, we conducted a Kaplan–Meier survival analysis to estimate the cumulative probability of remaining free from AF recurrence over time. This method allowed us to account for the timing of recurrences and provided a more nuanced understanding of the long-term efficacy of rhythm control strategies. Patients who were lost to follow-up or who had not experienced a recurrence by the end of the study period were censored in the analysis, and their contributions to the survival curve were appropriately adjusted. The Kaplan–Meier curve was generated to visually represent the survival function, with survival defined as the absence of AF recurrence. The curve provided insights into the durability of the treatment over the follow-up period and highlighted the times at which recurrences occurred, offering a comprehensive view of the rhythm control strategies’ effectiveness in our paediatric cohort.

A formal power analysis was not performed prior to this study due to the retrospective nature and the relatively rare occurrence of paediatric atrial fibrillation (AF). The study was designed to include all available cases within the specified time frame (January 2015 to December 2023) at our tertiary care centre. Given the rarity of AF in the paediatric population, the sample size was determined by the number of cases that met the inclusion criteria during this period. While we acknowledge that the lack of a prospective power analysis may limit the ability to detect small differences or outcomes, the study’s primary objective was to provide a comprehensive analysis of paediatric AF cases in our centre. The findings contribute valuable insights into this rare condition, despite the inherent limitations related to sample size.

## 3. Results

### 3.1. Demographics and Clinical Presentation

The study cohort included 36 paediatric patients diagnosed with atrial fibrillation (AF) between January 2015 and December 2023. The median age at presentation was 15 years (IQR: 13–17 years), with patients ranging from 10 to 18 years old. The majority of the patients were male (58%; *n* = 21). Nearly half of the AF diagnoses (47.2%; *n* = 17) were made in the emergency department, where patients primarily presented with symptoms such as tachycardia and palpitations.

A significant proportion of the cohort (86.1%; *n* = 31) had underlying cardiovascular conditions. The most common was acquired heart disease, particularly rheumatic heart disease (RHD), which was present in 52.8% (*n* = 19) of patients. Structural congenital heart disease was observed in 16.7% (*n* = 6), while rhythm disturbances, such as Wolff–Parkinson–White syndrome, were present in 8.3% (*n* = 3). A smaller subset of patients was diagnosed with dysautonomia (2.8%; *n* = 1) and cardiomyopathy (2.8%; *n* = 1). Notably, 16.7% (*n* = 6) of patients presented with lone atrial fibrillation, defined as AF without any identifiable structural or functional heart disease.

Among patients with RHD, isolated mitral valve involvement, predominantly stenosis, was observed in 57.9% (*n* = 11). Of these, seven patients underwent mechanical mitral valve replacement. Rheumatic mitro-aortic disease, characterised by stenosis, was present in seven patients, with four undergoing surgery that included mechanical mitral valve replacement and aortic valve plasty.

The cohort also included six patients with congenital heart anomalies. These included complete atrioventricular septal defect (*n* = 2), pulmonary valve atresia with ventricular septal defect (*n* = 1), univentricular heart with double-inlet left ventricle and malposition of the great arteries (*n* = 1), truncus arteriosus type I (*n* = 1), and a combination of atrial and septal defects with patent ductus arteriosus in one patient diagnosed with Shprintzen–Goldberg syndrome. All patients with congenital heart defects had undergone surgical correction.

Regarding the context of diagnosis, AF was identified during emergency visits in 47.2% (*n* = 17) of cases, during cardiac catheterisation in 5.6% (*n* = 2), post-surgically in 11.1% (*n* = 4), and as an incidental finding in 36.1% (*n* = 13). None of the patients had a family history of AF.

Detailed demographic and clinical characteristics of the cohort are summarised in Table 1.

### 3.2. Treatment Strategies

At initial presentation, 41.7% (*n* = 15) of the cohort required immediate intervention to manage atrial fibrillation (AF), while 27.8% (*n* = 10) spontaneously reverted to sinus rhythm. The primary interventions included electrical cardioversion (53.3%; *n* = 8) and pharmacological conversion with amiodarone (46.7%; *n* = 7). In 30.6% (*n* = 11) of cases, the specific treatment modality was not documented (Table 2).

Following the initial AF episode, 80.6% (*n* = 29) of patients were prescribed pharmacological therapy. Monotherapy was the most common approach, used in 69% (*n* = 20) of cases. Amiodarone (25%) and digoxin (22%) were the preferred agents. Oral anticoagulation was initiated in 63.9% (*n* = 23) of the cohort, particularly among those with rheumatic heart disease (RHD) (*n* = 20) and congenital structural heart disease (*n* = 3). Warfarin was the most frequently prescribed anticoagulant, while rivaroxaban was used in two cases of lone AF.

All patients with Wolff–Parkinson–White (WPW) syndrome or other accessory pathways underwent successful catheter ablation as the first-line treatment. Ablations, performed using either radiofrequency or cryoablation techniques, were highly effective, eliminating the need for further antiarrhythmic therapy. The ablation targeted various accessory pathways, including lateral left, posteroseptal, and mid-septal pathways, resulting in the complete resolution of WPW-associated arrhythmias.

Patients with lone AF underwent electrophysiological studies followed by catheter ablation. The ablation strategy primarily involved antral isolation of the pulmonary veins. Radiofrequency ablation was employed in 40% of these cases, while the remaining patients underwent single-shot cryoablation. Notably, two patients exhibited extensive areas of low voltage in the left atrium, prompting further genetic investigations, though these remained inconclusive. In one patient, arrhythmogenic triggers were identified and successfully ablated from the superior vena cava.

Before ablation procedures, all patients underwent a preprocedural transthoracic echocardiogram to assess left ventricular ejection fraction and left atrial dimensions. Computed tomography (CT) with left atrial segmentation was performed to evaluate atrial anatomy and exclude the presence of intracardiac thrombi. Oral anticoagulation was maintained until the procedure, with warfarin or direct oral anticoagulants (DOACs) administered at therapeutic doses, except for one dosage omitted before ablation. Antiarrhythmic drugs were discontinued for at least five half-lives before the procedure. Ablations were performed under conscious sedation with continuous monitoring of oxygen saturation and ECG.

The ablation protocol varied depending on the mapping system used. For patients treated with the CARTO system (Biosense Webster, Irvine, CA, USA), ablations were performed using a ThermoCool SmartTouch^®^ SurroundFlow catheter. For those treated with the NavX Precision system (Abbott, Abbott Park, IL, USA), the FlexAbility Ablation Catheter, Sensor Enabled, was used. In patients who underwent cryoablation, the Arctic Front™ cryoablation system (Medtronic, Minneapolis, MN) was used in conjunction with the Achieve Mapping Catheter (Medtronic) for precise mapping and ablation.

The ablation procedure involved several key steps: positioning a decapolar catheter through the right femoral vein to the coronary sinus for guidance, performing a transseptal puncture under fluoroscopic guidance, and employing three-dimensional mapping systems (either CARTO or NavX Precision) to map both the right and left atria. Ablations were then performed using point-by-point lesions to achieve antral isolation of the pulmonary veins. In cases where cryoablation was used, single-shot techniques were employed to isolate the pulmonary veins.

### 3.3. Recurrence

The overall efficacy of the rhythm control strategies in our cohort was 96.2%, with a recurrence rate of atrial fibrillation (AF) of 3.8% (*n* = 3) over the follow-up period. To provide a more precise assessment of the timing and likelihood of recurrence, we conducted a Kaplan–Meier survival analysis (Figure 1). The median follow-up time was 30 months [IQR: 24–36 months].

Recurrences were documented in three patients. Two of these patients had lone AF—one experienced a recurrence while on beta-blocker therapy and the other while on amiodarone. The third recurrence occurred in a patient with congenital structural heart disease (Truncus arteriosus type I), who was being treated with a combination of a beta-blocker and digoxin.

### 3.4. Outcomes and Complications

During the follow-up period, we experienced a significant loss of data, with 19 patients—mostly those with rheumatic heart disease (RHD) and from Portuguese-speaking African countries—being lost to follow-up after returning to their home countries. This loss is concerning, as it introduces potential bias, especially given the specific demographic affected. The absence of follow-up data from these patients could distort the study’s findings, particularly regarding long-term outcomes and the effectiveness of rhythm control strategies in patients with RHD.

Ischaemic stroke was the most common complication, occurring in three patients, all of whom had underlying RHD. One patient had unoperated left atrial dilation, another had a mitral prosthesis, and the third had undergone mitral valvuloplasty but was noncompliant with anticoagulation therapy. These strokes resulted in significant residual deficits, including hemiparesis in two patients and speech difficulties in one. These outcomes underscore the critical importance of consistent follow-up and strict adherence to anticoagulation therapy to prevent such severe complications, particularly in patients with RHD.

In addition to strokes, the management of anticoagulation therapy posed significant challenges. Although no major bleeding events were reported in this cohort, several patients, particularly those with RHD, struggled to maintain therapeutic levels of anticoagulation. Fluctuations in the International Normalised Ratio (INR) levels were common, with some patients experiencing dangerously elevated INR levels, significantly increasing their risk of bleeding. Minor bleeding episodes were reported in four patients, with two cases of epistaxis and two cases of gingival bleeding. These incidents highlight the delicate balance required in managing anticoagulation therapy.

These findings emphasise the importance of rigorous monitoring and management of anticoagulation, particularly in patients with mechanical heart valves or those at high risk of thromboembolism. Tailored strategies are needed to minimise both thrombotic and bleeding complications, ensuring optimal outcomes for these high-risk patients.

## 4. Discussion

Atrial fibrillation (AF) is the most common arrhythmia in adults, closely linked to risk factors like hypertension, valvular heart disease, and ageing [8,9]. However, AF is rare in children, with a prevalence of less than 0.05% before the age of 30 [2]. In paediatric cases, AF is often associated with underlying conditions such as congenital heart disease (CHD), rheumatic heart disease (RHD), cardiomyopathy, and inherited arrhythmias [1,5]. The rarity of AF in children, along with its connection to congenital and acquired heart conditions, highlights the need for focused research to better understand and treat AF in this age group. This study adds to the existing literature by providing a detailed analysis of AF in children, examining its epidemiology, underlying conditions, management strategies, and outcomes.

### 4.1. AF and Underlying Conditions

Our findings align with previous studies, showing that AF in children is often linked to underlying cardiovascular conditions. In our cohort, 86.1% of the patients had an associated cardiovascular condition, with acquired heart disease, particularly rheumatic heart disease (RHD), being the most common (52.8%). This supports earlier reports that identified RHD as a leading cause of AF in paediatric populations, especially in areas with high rates of rheumatic fever [10]. Although Portugal is not typically a high-burden country for RHD, it receives many migrants from regions where RHD is still prevalent. Our findings emphasise the importance of recognising and managing RHD-related AF in non-endemic regions. This highlights the need for preventive measures against rheumatic fever to reduce the burden of AF in affected children, particularly those from underserved or migrant populations.

Consistent with other studies, we also found that congenital heart disease (CHD) significantly contributes to AF in children [3,11]. Congenital defects, such as atrial septal defects and more complex anomalies like truncus arteriosus, were present in 16.7% of our patients. Mandalenakis et al. demonstrated that children with CHD have a 22-fold increased risk of developing AF compared to the general population, even though the absolute risk remains low during childhood [12]. Our findings reinforce the need for long-term surveillance in children with CHD, as AF may appear earlier in life, especially after surgical interventions that alter the atrial anatomy [13]. This is crucial in managing children with complex congenital anomalies, where surgical corrections can create new arrhythmogenic substrates, increasing the risk of AF.

### 4.2. WPW Syndrome and Accessory Pathways

Our study found that 8.3% of the patients had Wolff–Parkinson–White (WPW) syndrome, with atrial fibrillation (AF) being a common presenting arrhythmia. The literature suggests that 16–26% of patients with WPW may experience spontaneous degeneration of atrioventricular re-entrant tachycardia (AVRT) into AF, which is dangerous due to the risk of rapid ventricular responses and potential progression to ventricular fibrillation [14]. In our cohort, all patients with WPW underwent successful catheter ablation of their accessory pathways. This intervention led to the resolution of AF and eliminated the need for ongoing antiarrhythmic medication. These results align with other studies that recommend catheter ablation as the first-line treatment for WPW-associated AF to prevent life-threatening arrhythmias and improve long-term outcomes [14].

### 4.3. Lone Atrial Fibrillation

We observed a higher prevalence of lone AF in our paediatric cohort (16.7%) compared to previously reported data, such as the study by El-Assaad et al., which indicated a prevalence of 7.5 per 100,000 children [5]. This discrepancy may be partly due to the size of our study population and specific geo-social factors relevant to our cohort. Lone AF, characterised by the absence of identifiable structural heart disease or other conditions, presents a significant diagnostic and therapeutic challenge. Recent studies suggest that atrial fibrosis may play a crucial role in the pathogenesis of AF, even in young patients [7].

In our study, two patients with lone AF showed extensive areas of low voltage in the left atrium, indicating underlying atrial fibrosis. One of these patients, who had no family history of arrhythmia or cardiomyopathy, underwent cardiac MRI, which revealed isolated late gadolinium enhancement of the posterior wall of the left atrium. This finding is similar to those previously described in patients with MYH7 variants associated with extensive left atrial fibrosis in the context of paroxysmal AF [6]. Although genetic testing in our patients did not yield definitive results, these findings support the hypothesis that lone AF may represent an early manifestation of atrial-selective cardiomyopathy, where the primary pathology resides within the atria rather than being secondary to other cardiac conditions [15].

Emerging evidence suggests that genetic variants, particularly those affecting the sodium and potassium channels, cardiomyopathy-related genes (such as TTN), gap junction channels, and transcription factors, may contribute to the development of AF in younger populations [6,16]. These findings highlight the importance of a comprehensive evaluation, including advanced imaging and genetic testing, in paediatric patients presenting with lone AF. Such evaluations are essential to better understand the underlying mechanisms and to guide appropriate management.

### 4.4. The Role of the Autonomic Nervous System in Atrial Fibrillation

In addition to structural and genetic factors, the autonomic nervous system (ANS) is increasingly recognised as playing a pivotal role in the pathogenesis of atrial fibrillation. Dysautonomia, characterised by an imbalance or dysfunction in the autonomic regulation of the heart, has been implicated in the initiation and maintenance of AF. The ANS influences atrial electrophysiology through its sympathetic and parasympathetic branches, which can alter atrial refractoriness and conduction velocity, thereby creating a substrate conducive to AF [17,18].

In our cohort, we observed one patient diagnosed with dysautonomia, a condition that may contribute to AF through heightened sympathetic activity or vagal tone. Studies have shown that increased sympathetic activity can lead to atrial ectopy and shortens the atrial refractory periods, while an increased vagal tone can promote re-entrant circuits by slowing conduction [19]. This autonomic imbalance may serve as a trigger for AF episodes, particularly in patients without structural heart disease. Understanding the role of the ANS in AF, especially in younger patients with conditions like dysautonomia, could help refine therapeutic strategies, such as targeting autonomic modulation through pharmacological or interventional approaches.

Given the complex interplay between the autonomic nervous system and AF, further research is warranted to explore how dysautonomia and related autonomic dysfunctions might contribute to the pathogenesis of AF in paediatric populations. This could lead to more targeted therapies aimed at autonomic regulation, potentially reducing the AF burden in these patients.

### 4.5. Management and Outcomes

In our study, the most commonly used initial interventions for managing atrial fibrillation (AF) in paediatric patients were electrical cardioversion (53.3%) and amiodarone therapy (46.7%). These approaches are aligned with the current recommendations for the acute management of AF in paediatric populations. The notably low recurrence rate observed in our cohort (3.8%) is particularly significant, especially when compared to higher recurrence rates reported in other studies. This lower recurrence rate may be attributed to the high proportion of patients in our cohort who underwent catheter ablation, particularly those with Wolff–Parkinson–White (WPW) syndrome, where ablation effectively eliminated the substrate for AF recurrence.

Oral anticoagulation was initiated in 63.9% of the patients, primarily those with rheumatic heart disease (RHD) and congenital heart defects. Warfarin was the most commonly prescribed anticoagulant in our cohort, reflecting the specific requirements of patients with conditions such as mechanical valve replacements due to RHD. However, the occurrence of ischemic stroke in three patients with RHD, despite anticoagulation therapy, underscores the challenges in managing stroke risk within this high-risk group. This finding is consistent with the existing literature, which emphasises the critical role of anticoagulation in preventing thromboembolic events in children with AF and underlying heart disease.

Moreover, the management of anticoagulation in our cohort posed challenges, with some patients experiencing difficulties in maintaining therapeutic International Normalised Ratio (INR) levels, leading to episodes of suboptimal anticoagulation. Although no major bleeding events were reported, several patients experienced minor bleeding complications, such as epistaxis and gingival bleeding. These incidents highlight the delicate balance required in managing anticoagulation therapy, particularly in paediatric patients who are at an increased risk of both thromboembolic and haemorrhagic complications.

The high rate of data loss in our cohort, primarily due to patients returning to their home countries, limits our ability to fully assess long-term outcomes and complications. This limitation is particularly concerning, because it introduces the potential for bias, especially given that the missing data predominantly pertain to patients with RHD from Portuguese-speaking African countries. The absence of follow-up data for these patients may skew the study’s findings, particularly concerning the long-term efficacy of rhythm control strategies and the true incidence of complications. Therefore, there is a pressing need for more comprehensive and consistent follow-up strategies, particularly in populations that are difficult to track due to geographic or socioeconomic factors. This would ensure a more accurate assessment of long-term outcomes and a better understanding of the effectiveness of various treatment strategies in paediatric AF.

### 4.6. Safety Profile and Atrial Stiffness

In our study, we primarily used radiofrequency and cryoablation for catheter ablation procedures in paediatric patients with AF. While these methods are effective, they are not without risks, particularly concerning the potential for atrial stiffness and other complications. Recent research has explored the impact of these traditional ablation techniques on atrial tissue, with studies suggesting that atrial stiffness might contribute to adverse outcomes, especially in patients with existing comorbidities like pulmonary hypertension.

Pulsed-field ablation (PFA) represents an exciting advancement in the treatment of atrial fibrillation. Unlike conventional radiofrequency and cryoablation techniques, which rely on thermal energy to destroy arrhythmogenic tissue, PFA uses non-thermal, high-voltage electrical fields to selectively target and ablate cardiac tissue. This approach minimises collateral damage to the surrounding structures, such as the oesophagus, phrenic nerve, and pulmonary veins, which is particularly important in the paediatric population.

Recent studies have highlighted the effectiveness of PFA in achieving durable pulmonary vein isolation with a reduced risk of complications. For instance, a review published by Pierucci et al. discussed the clinical applications and benefits of PFA in the treatment of AF, noting its potential to improve outcomes while reducing the risk of procedural complications [20].

Additionally, emerging evidence indicates that PFA may offer a safer alternative. According to a study by Mohanty et al., PFA does not worsen baseline pulmonary hypertension following prior radiofrequency ablations, suggesting a favourable safety profile for PFA in patients at risk of atrial stiffness [21]. This is particularly relevant in paediatric patients, who may benefit from less invasive and tissue-sparing techniques.

The precision and safety profile of PFA could be particularly advantageous for younger patients who may require long-term management of their arrhythmias. As such, incorporating PFA into treatment protocols for paediatric AF could represent a significant step forward in improving patient outcomes.

### 4.7. Future Directions and Clinical Implications

Our study highlights several important considerations in the management of paediatric AF. First, the high prevalence of AF in children with RHD and CHD underscores the need for early detection and targeted intervention in these high-risk groups. Second, the success of catheter ablation in managing WPW-associated AF suggests that early intervention may be beneficial for preventing recurrence and improving long-term outcomes. Finally, the emerging recognition of lone AF as a potential manifestation of early atrial cardiomyopathy calls for further research on the genetic and molecular mechanisms underlying AF in young patients.

Larger multicentre studies with robust follow-up are essential to better understand the natural history of paediatric AF and optimise treatment strategies. The integration of advanced imaging techniques, such as cardiac MRI and genetic testing, may provide further insights into the pathophysiology of AF in this population, paving the way for personalised therapeutic approaches. Clinically, our study underscores the importance of early and comprehensive management of paediatric AF to prevent complications and improve long-term outcomes in this vulnerable patient group. Addressing the challenges associated with follow-up and data retention will be crucial for future research efforts to ensure that the findings are generalisable and that paediatric AF management continues to evolve based on robust evidence.

### 4.8. Limitations

This study has several limitations that need to be considered. First, the retrospective design, conducted at a single tertiary paediatric cardiology centre, limits our ability to establish causation between observed factors and outcomes. The retrospective nature also introduces potential biases, such as selection bias and the impact of incomplete medical records, which may affect the accuracy and reliability of the findings. Additionally, the absence of a prospective power analysis means that we cannot be certain if the sample size was sufficient to detect significant differences or outcomes, which may compromise the robustness of our results.

The small sample size and the significant loss of follow-up data from 19 patients—mainly those from Portuguese-speaking African countries with rheumatic heart disease (RHD)—are also concerning. This loss could introduce bias, as these patients’ outcomes might differ from those who remained in the study, potentially skewing the results, particularly regarding long-term outcomes and the effectiveness of rhythm control strategies in patients with RHD.

The heterogeneity in the treatment protocols across the cohort is another limitation. Differences in how patients were managed, especially regarding anticoagulation therapy and ablation techniques, may have influenced the results. This variability, along with the small sample size, prevented us from performing a detailed comparison of outcomes across different treatment strategies, as suggested by a reviewer.

These limitations mean that our findings should be interpreted with caution. Larger, multicentre studies with prospective designs, consistent follow-up, and standardised treatment protocols are needed to confirm these results and to better understand the optimal management of paediatric atrial fibrillation.

## 5. Conclusions

In our study population, AF was predominantly associated with underlying structural heart disease; however, nearly 15% of patients presented with lone AF, indicating that AF can occur without a prior cardiac substrate. Although AF has a high rate of conversion to sinus rhythm, most patients require long-term antiarrhythmic therapy, reflecting the chronic nature of the condition. Neurological complications, particularly ischaemic strokes, were a significant concern but were limited to patients with underlying heart disease, emphasising the importance of careful management in this subgroup. Further, multicentre studies with larger patient cohorts are essential to better understand the aetiology, risk factors, and optimal therapeutic interventions for paediatric AF. These future studies could provide more comprehensive data to guide clinical practice and improve outcomes in children affected by this complex arrhythmia.

## Figures and Tables

**Figure 1 medicina-60-01505-f001:**
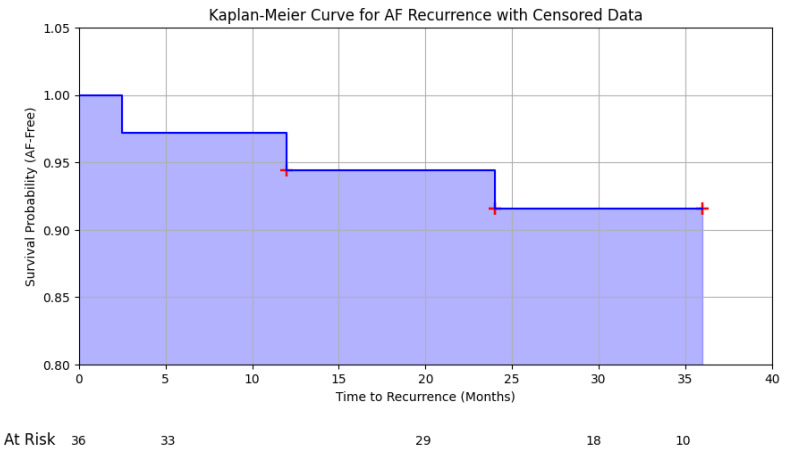
Kaplan-Meier survival curve illustrating the cumulative probability of the remaining free from atrial fibrillation (AF) recurrence over the follow-up period in a cohort of 36 paediatric patients. The curve represents the survival function, with survival defined as the absence of AF recurrence. The median follow-up time was 30 months [IQR: 24–36 months]. The drops in the curve correspond to the three documented recurrences at 2.5, 12, and 24 months. The number of patients at risk at each key time point is indicated below the x-axis. The analysis demonstrates a high overall efficacy of rhythm control strategies, with a recurrence rate of 3.8% over the study period.

**Table 1 medicina-60-01505-t001:** Patient demographics and diagnosis context.

Variable	*n* = 36
Baseline characteristics	
Age (y), median	15
Male sex, n (%)	21 (58%)
Coexisting conditions	
Structural Congenital HD	6 (16.7%)
Structural Acquired HD	19 (52.8%)
Cardiomyopathy	1 (2.8%)
Rhythm Diseases	3 (8.3%)
Dysautonomia	1 (2.8%)
Without known cardiac disease	6 (16.7%)
Family history of AF, n (%)	0 (0%)
Context of diagnosis	
Emergency, n (%)	17 (47.2%)
During cardiac catheterisation, n (%)	2 (5.6%)
Post-surgical, n (%)	4 (11.1%)
Incidental finding, n (%)	13 (36.1%)

**Table 2 medicina-60-01505-t002:** Treatment strategies.

Therapy	n	%
**Initial approach**		
Unknown	11	30.6
Needed intervention	15	41.7
Amiodarone	7	46.7
External electric cardioversion	8	53.3
Spontaneous resolution	10	27.8
**Long-term approach**		
Pharmacological treatment	29	80.6
Monotherapy	20	69
Digoxin	8	22
Amiodarone	9	25
Flecainide	2	6
Sotalol	1	3
Multiple	9	31
Amiodarone + digoxin	5	14
Flecainide + beta-blocker	4	11
Without pharmacological treatment	7	19

## Data Availability

The data supporting the reported results in this study are not publicly available due to privacy and ethical restrictions. The study involved patient medical records, and the data were anonymised to protect patient confidentiality in accordance with institutional guidelines. Access to the data is restricted and may be available from the corresponding author upon reasonable request, subject to approval by the Institutional Review Board and in compliance with applicable data protection regulations.

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
