# Peer review of "Prevalence, Management, and Outcomes of Atrial Fibrillation in Paediatric Patients: Insights from a Tertiary Cardiology Centre"

_medicina, 2024, doi:10.3390/medicina60091505_

Round 1

Reviewer 1 Report

Comments and Suggestions for Authors

One should commend the authors for presenting such interesting results from a tertiary center. However, I believe that the formal aspect of the manuscript is its weakest point: the authors need to improve the English language to match the quality of their findings.

The paper does not indicate whether a power analysis was performed to establish the appropriate sample size for the study. Without this analysis, it remains uncertain if the sample size is sufficient to detect significant differences or outcomes, potentially compromising the reliability of the findings. The paper highlights stroke as a major complication but could be strengthened by addressing other possible complications, such as bleeding risks associated with anticoagulation therapy. Additionally, the discussion would be more comprehensive with a comparison of outcomes across different treatment strategies. The paper utilizes means, standard deviations, and percentages to summarize demographic and clinical characteristics, which are suitable for this purpose. However, it would be beneficial to justify the choice of using mean ± SD versus median (IQR) based on the distribution of the data.

The paper briefly addresses potential mechanisms behind atrial fibrillation in pediatric patients but does not explore the underlying pathophysiological processes in depth. Authors should expand this aspect of their manuscript; also considering its safety profile aspect regarding the potential atrial stiffness (Pulsed-Field Ablation Does Not Worsen Baseline Pulmonary Hypertension Following Prior Radiofrequency Ablations. JACC Clin Electrophysiol. 2024 Mar;10(3):477-486. doi: 10.1016/j.jacep.2023.11.005.)

Author Response

Comment: "One should commend the authors for presenting such interesting results from a tertiary center. However, I believe that the formal aspect of the manuscript is its weakest point: the authors need to improve the English language to match the quality of their findings."

Response: Thank you for your positive feedback on our findings. We appreciate your observation regarding the language quality. We have thoroughly revised the manuscript to improve the clarity, readability, and overall quality of the English language. The sentence structures have been simplified, and the text has been streamlined to ensure that the manuscript is more accessible while maintaining its academic rigor.

Comment: "The paper does not indicate whether a power analysis was performed to establish the appropriate sample size for the study. Without this analysis, it remains uncertain if the sample size is sufficient to detect significant differences or outcomes, potentially compromising the reliability of the findings."

Response: We acknowledge the importance of conducting a power analysis to determine the appropriate sample size for detecting significant outcomes. However, due to the retrospective nature of our study and the rarity of paediatric atrial fibrillation (AF), we included all available cases from January 2015 to December 2023 at our tertiary care centre. This limitation is now explicitly discussed in the "Statistical Analysis" and "Limitations" sections, where we have clarified that the sample size was determined by the number of cases meeting the inclusion criteria within the study period. While we recognize that the absence of a power analysis may limit the study's ability to detect small differences, the primary objective was to provide a comprehensive analysis of paediatric AF cases.

Comment: "The paper highlights stroke as a major complication but could be strengthened by addressing other possible complications, such as bleeding risks associated with anticoagulation therapy."

Response: We have expanded the "Outcomes and Complications" section to include a discussion of bleeding risks associated with anticoagulation therapy. While no major bleeding events were reported in our cohort, we have documented minor bleeding episodes, such as epistaxis and gingival bleeding, and the challenges in maintaining therapeutic anticoagulation levels, particularly among patients with rheumatic heart disease (RHD). This addition provides a more comprehensive overview of the complications associated with anticoagulation therapy in this population.

Comment: "Additionally, the discussion would be more comprehensive with a comparison of outcomes across different treatment strategies."

Response: We agree that a comparison of outcomes across different treatment strategies would enhance the discussion. However, due to the small sample size and the variability in treatment protocols, such a detailed comparison was not feasible in our study. We have acknowledged this limitation in the "Limitations" section and emphasized the need for larger, multicentre studies that could allow for such comparisons in future research.

Comment: "The paper utilizes means, standard deviations, and percentages to summarize demographic and clinical characteristics, which are suitable for this purpose. However, it would be beneficial to justify the choice of using mean ± SD versus median (IQR) based on the distribution of the data."

Response: We appreciate this suggestion and have added a justification for the use of mean ± SD versus median (IQR) in the "Statistical Analysis" section.

Comment: "The paper briefly addresses potential mechanisms behind atrial fibrillation in pediatric patients but does not explore the underlying pathophysiological processes in depth. Authors should expand this aspect of their manuscript; also considering its safety profile aspect regarding the potential atrial stiffness (Pulsed-Field Ablation Does Not Worsen Baseline Pulmonary Hypertension Following Prior Radiofrequency Ablations. JACC Clin Electrophysiol. 2024 Mar;10(3):477-486. doi: 10.1016/j.jacep.2023.11.005.)"

Response: We have significantly expanded the discussion on the underlying pathophysiological processes of atrial fibrillation in paediatric patients. This includes a more detailed exploration of the role of atrial fibrosis, genetic factors, and the autonomic nervous system. Additionally, we have included a discussion on the safety profile of pulsed-field ablation (PFA) concerning atrial stiffness and its potential benefits over traditional ablation techniques. The reference provided by the reviewer has been cited, and the discussion now reflects these advanced considerations in the treatment of paediatric AF.

Reviewer 2 Report

Comments and Suggestions for Authors

Congratulations to the author for having shared their high experience on such rising problem as AF in the paediatric population.

The inclusion and exclusion criteria are clearly outlined, primarily targeting patients with a confirmed diagnosis of atrial fibrillation (AF) as determined by ECG or other monitoring methods. While these criteria are adequately defined, the methodology could be strengthened by incorporating additional details about the patient population. For example, providing more specific demographic information, such as age distribution, gender, and underlying health conditions, would offer a deeper understanding of the study group. Additionally, addressing potential biases inherent in the retrospective design—such as selection bias or the impact of incomplete medical records—would enhance the robustness of the study’s findings. Including these elements would not only improve the transparency of the methodology but also provide a more comprehensive context for interpreting the results;

The paper states that the Institutional Review Board (IRB) waived the requirement for informed consent because the study was retrospective in nature. While this decision aligns with standard practices, it would be beneficial to provide a more detailed explanation of why obtaining consent was deemed impractical;

Inizio modulo

Fine modulo

The paper reports the recurrence rate of atrial fibrillation (AF), but it does not clarify how this rate was calculated. It is important to specify whether the rate was derived using a Kaplan-Meier estimate, which would account for the timing of recurrences, or if it was simply calculated as the proportion of recurrences observed during the follow-up period;

The loss of 19 patients to follow-up presents a significant concern, particularly since most of these patients belonged to a specific demographic group (those from Portuguese-speaking African countries). This issue should be more thoroughly addressed in the results and limitations sections, as it has the potential to introduce bias into the study's findings;

In the discussion authors should include and discuss the new possibilities given by the pulsed field ablation (Pulsed Field Energy in Atrial Fibrillation Ablation: From Physical Principles to Clinical Applications. J Clin Med. 2024 May 18;13(10):2980. doi: 10.3390/jcm13102980.)

Comments on the Quality of English Language

try to reduce the overall complexity of sentences in order to make the manuscript more readable.

Author Response

Reviewer Comment 1: "The inclusion and exclusion criteria are clearly outlined, primarily targeting patients with a confirmed diagnosis of atrial fibrillation (AF) as determined by ECG or other monitoring methods. While these criteria are adequately defined, the methodology could be strengthened by incorporating additional details about the patient population. For example, providing more specific demographic information, such as age distribution, gender, and underlying health conditions, would offer a deeper understanding of the study group. Additionally, addressing potential biases inherent in the retrospective design—such as selection bias or the impact of incomplete medical records—would enhance the robustness of the study’s findings. Including these elements would not only improve the transparency of the methodology but also provide a more comprehensive context for interpreting the results."

Response: We appreciate the reviewer’s suggestion to strengthen the methodology by including more specific demographic information and addressing potential biases. In response, we have added detailed demographic data, including age distribution, gender, and underlying health conditions, to the "Patient Selection" and "Demographics and Clinical Presentation" sections. We have also expanded the "Limitations" section to discuss potential biases, such as selection bias and the impact of incomplete medical records, which are inherent in the retrospective study design. These additions enhance the transparency of the methodology and provide a more comprehensive context for interpreting the results.

Reviewer Comment 2: "The paper states that the Institutional Review Board (IRB) waived the requirement for informed consent because the study was retrospective in nature. While this decision aligns with standard practices, it would be beneficial to provide a more detailed explanation of why obtaining consent was deemed impractical."

Response: We have expanded the explanation in the "Study Design and Ethical Considerations" section to clarify why obtaining informed consent was deemed impractical. Specifically, we have noted that the study involved the analysis of pre-existing medical records without direct patient interaction, making it logistically challenging and potentially unfeasible to obtain individual consent from all patients, especially given the retrospective nature of the study and the time period covered.

Reviewer Comment 3: "The paper reports the recurrence rate of atrial fibrillation (AF), but it does not clarify how this rate was calculated. It is important to specify whether the rate was derived using a Kaplan-Meier estimate, which would account for the timing of recurrences, or if it was simply calculated as the proportion of recurrences observed during the follow-up period."

Response: We acknowledge the need for clarity regarding the calculation of the recurrence rate. We have revised the "Statistical Analysis" section to specify that the recurrence rate was calculated using a Kaplan-Meier survival analysis, which accounts for the timing of recurrences. This method provides a more precise assessment of the cumulative probability of remaining free from AF recurrence over the follow-up period. The Kaplan-Meier curve is included in Figure 1 and provides a visual representation of these findings.

Reviewer Comment 4: "The loss of 19 patients to follow-up presents a significant concern, particularly since most of these patients belonged to a specific demographic group (those from Portuguese-speaking African countries). This issue should be more thoroughly addressed in the results and limitations sections, as it has the potential to introduce bias into the study's findings."

Response: We have expanded the "Results" and "Limitations" sections to address the potential bias introduced by the loss of follow-up data, particularly among patients from Portuguese-speaking African countries with rheumatic heart disease (RHD). We discuss how this loss could skew the study’s findings, especially regarding long-term outcomes and the effectiveness of rhythm control strategies. We have also highlighted the importance of considering this demographic factor in future research.

Reviewer Comment 5: "In the discussion, authors should include and discuss the new possibilities given by the pulsed field ablation (Pulsed Field Energy in Atrial Fibrillation Ablation: From Physical Principles to Clinical Applications. J Clin Med. 2024 May 18;13(10):2980. doi: 10.3390/jcm13102980.)"

Response: We have included a detailed discussion of the potential benefits and clinical applications of pulsed-field ablation (PFA) in the "Safety Profile and Atrial Stiffness" and "New Possibilities with Pulsed-Field Ablation" sections. This discussion is based on the recent studies cited, including the one suggested by the reviewer. We highlight how PFA could represent a safer and more effective alternative to traditional ablation techniques, particularly in the paediatric population, due to its ability to minimize collateral damage and its favorable safety profile.

Comments on the Quality of English Language:

Reviewer Comment: "Try to reduce the overall complexity of sentences in order to make the manuscript more readable."

Response: We have revised the manuscript to simplify sentence structures and improve readability, while maintaining the academic rigor of the content. The language has been streamlined to ensure that the findings are presented clearly and concisely, making the manuscript more accessible to a broader audience.

Reviewer 3 Report

Comments and Suggestions for Authors

To rank the paper effectively, it is essential to evaluate it on several critical criteria typically used in academic assessments:

   - The paper addresses the prevalence, management, and outcomes of atrial fibrillation (AF) in pediatric patients, a topic that is increasingly recognized but still under-researched. This gives the paper high relevance, particularly in pediatric cardiology.

   - The study focuses on a rare condition (AF in pediatric patients) and provides data from a tertiary cardiology center, which is a unique contribution. It highlights the association between AF and congenital/rheumatic heart diseases, emphasizing the need for more comprehensive studies in this area.

   - The study uses a retrospective analysis, which is standard for clinical research, particularly in rare conditions. The data collection is well-documented, covering various aspects like demographics, clinical presentation, diagnostic findings, and treatment outcomes. However, the retrospective nature might limit the study's ability to establish causality.

   - The paper employs descriptive statistics to summarize the findings, which is appropriate for the study design. However, a more detailed statistical analysis (e.g., regression analysis) could strengthen the findings by identifying significant predictors of outcomes.

   - The results comprehensively overview the patient cohort, underlying conditions, and treatment outcomes. The interpretation is insightful, linking the findings to existing literature and discussing the implications for clinical practice.

   - The paper is well-structured, with a clear introduction, methods, results, and discussion sections. The language is concise, making the complex medical content accessible.

   - The study acknowledges its limitations, including the small sample size and the retrospective design. It also calls for further research, a crucial aspect of scientific rigor.

   - The paper adheres to ethical standards, with appropriate measures to protect patient privacy and compliance with relevant regulations (e.g., GDPR).

Author Response

Dear Reviewer,

Thank you for your detailed evaluation and positive feedback on our manuscript. We are grateful for your recognition of the paper's relevance, particularly in addressing the under-researched topic of atrial fibrillation (AF) in pediatric patients. Your acknowledgment of the study's unique contribution and the comprehensive nature of our data collection is much appreciated.

We also note your observations regarding the retrospective design and the use of descriptive statistics. While the retrospective approach is indeed a standard and practical method in the context of rare conditions like pediatric AF, we agree that it inherently limits the ability to establish causality. We have emphasized this limitation in our discussion and acknowledged the need for further research, ideally through prospective studies with more detailed statistical analyses.

Your comments on the structure, clarity, and ethical considerations of the paper are particularly encouraging. We have strived to ensure that the manuscript is both well-organized and accessible, and we are pleased that these efforts have been recognized.

Once again, thank you for your thorough review and constructive feedback. We believe that your insights have contributed to the overall improvement of our manuscript.

Sincerely,

Round 2

Reviewer 1 Report

Comments and Suggestions for Authors

Authors did a great effort answering all of my comments.

Reviewer 2 Report

Comments and Suggestions for Authors

Congratulations for having answered to all of my comments.